

# Creation of complex reef structures through coral restoration does not affect associated fish populations on a remote, well-protected, Caribbean reef

Jack V. Johnson[1], John F. Bruno[2], Lucas Le Gall[1], Matthew Louis Doherty[1], Alex Chequer[1] and Gretchen Goodbody Gringley[1]

[1] Reef Ecology and Evolution Lab, Central Caribbean Marine Institute, Little Cayman, Cayman Islands
[2] Department of Biology, University of North Carolina at Chapel Hill, Chapel Hill, United States of America

## ABSTRACT

Coral reef ecosystems are facing severe degradation due to anthropogenic activities at both local and global scales. In response, extensive restoration efforts are underway, aiming to bolster coral cover and enhance reef fish communities to foster facilitation between fish and corals. This reciprocal relationship is anticipated to improve overall restoration efficacy and enhance coral reef resilience in the face of global warming. Here, we investigate the impact of coral restoration using out-planted *Acropora cervicornis* colonies attached to raised domes on the associated fish community on the isolated, well-protected reef of Little Cayman Island in the Central Caribbean. Surveys were conducted immediately preceding out-planting, five days later, and 85 days later to capture temporal changes in the fish community. After 85 days of out-planting, there were no changes in fish biomass, abundance, or species richness for the entire fish community. This pattern was consistent for selected fish functional groups. Additionally, no significant differences were observed in the fish community before outplanting, five days after out-planting, or 85 days after out-planting of restoration domes. Our results underscore the limited impact of coral restoration for influencing fish communities in the isolated and highly protected reef of Little Cayman over an 85-day period. Consequently, our findings have implications for using coral restoration as a mechanism to enhance fish populations, particularly in marginally disturbed regions where structural complexity has not been lost. Future restoration programs should therefore incorporate local knowledge of environmental history and restoration needs along with an increased data-driven understanding of the intricate interaction between fish and coral populations to be successful.

# INTRODUCTION

Coral reefs are among the most productive marine ecosystems, providing crucial ecological services to sustain human livelihoods throughout the tropics (*Moberg & Folke, 1999*; *Woodhead et al., 2019*). However, coral reefs are threatened by a plethora of anthropogenic

Corresponding author
Jack V. Johnson,
jackvjohnson@hotmail.com

activity both at the local and global scale (*Hughes et al., 2017*). Given the extraordinarily high economic value of coral reefs, and their essential role for maintaining human livelihoods for up to one billion people, the UN nominated "Decade on Ecosystem Restoration of 2021 to 2030" (*Fischer et al., 2021*) places coral reef restoration as a high priority agenda to safeguard valuable ecosystem services (*Duarte et al., 2020*; *Hughes et al., 2023*; *Suggett et al., 2023*). Coral reef restoration ranges from small scale clearly defined restoration projects (*Fox et al., 2019*; *Ladd, Burkepile & Shantz, 2019*) to region-wide ambitious restoration efforts (*Gibbs et al., 2021*; *Hughes et al., 2023*). However, many reef restoration projects fail to establish tangible and ecologically relevant objectives (*Boström-Einarsson et al., 2020*; *Hughes et al., 2023*). Yet, certain ecologically relevant, and societally beneficial outcomes can be achieved if the purpose of restoration is clearly defined (*Boström-Einarsson et al., 2020*; *Lamont et al., 2022*; *Suggett et al., 2023*).

One of the key outcomes of coral reef restoration projects is to enhance the biomass, species richness, and functional diversity of reef fish communities, delivering multifaceted benefits to coral reef ecosystems (*Moberg & Folke, 1999*; *Woodhead et al., 2019*; *Eddy et al., 2021*). For example, algae grazing by herbivorous fish is necessary for coral fragments and transplants to not be outcompeted in their early life stages by macroalgae, thus influencing survival (*Edwards, 2010*; *Seraphim et al., 2020*). By enhancing the biomass of reef fish communities, key ecosystems function, such as energy flux through the system, can be maintained (*Oliver et al., 2015*; *Brandl et al., 2019*) even under multiple stressors (*Benkwitt, Wilson & Graham, 2020*). In particular, supporting key functional groups such as herbivorous fish reduces macroalgae dominance when disturbance events that destroy reef corals occur—ameliorating coral population recovery (*Hughes et al., 2007*). Therefore, by restoring reef-building corals to promote a functional fish community, coral reef resilience (*i.e.,* the resistance to and recovery from disturbance) should be enhanced (*Shaver & Silliman, 2017*; *Shaver et al., 2022*). This positive feedback-loop of coral restoration prompting functional fish groups (*i.e.,* facilitation) is a key facet for effective coral reef restoration techniques in the Anthropocene (*Seraphim et al., 2020*; *Boström-Einarsson et al., 2020*; *Shaver et al., 2022*; *Lamont et al., 2022*).

Despite the importance of reef fish for coral reef resilience, ecosystem function, and provision of ecosystem services, the influence of restoration on fish communities is often not reported—or reported with mixed effects (*Ladd et al., 2018*; *Seraphim et al., 2020*). For example, over a seven-month period in Florida, reef fish biomass and abundance significantly increased at restoration sites compared to control sites (*Opel et al., 2017*). In contrast, another short-term study (~2 months) found no effect of restoration on fish assemblages (*Ladd, Burkepile & Shantz, 2019*), while across multiple sites in the Caribbean no influence existed (*Huntington et al., 2017*) except from Dry Tortugas where facilitation did occur (*Huntington et al., 2017*). Additionally, a long-term study of over eight years showed restoration did not increase fish abundance or biomass at another site in Florida, or in the US Virgin Islands (*Hein et al., 2020*). Rather, the most common effect of restoration for influencing fish assemblages is increased Damselfish (Pomacentridae) abundance (*Merolla et al., 2013*; *Huntington et al., 2017*; *Ladd, Burkepile & Shantz, 2019*), which often
have negative impacts on coral restoration success by scarring coral tissue, and farming algae on coral outplants (*Quinn & Kojis, 2006*; *Forrester et al., 2012*; *Lohr et al., 2017*).

Given the dire state of coral reefs under global climate change and the mixed findings of restoration for influencing fish communities over the short term, reports from case studies are useful steppingstones to build a holistic inference on the efficacy of restoration for enhancing fish communities that could aid restoration (*Seraphim et al., 2020*; *Shaver et al., 2022*). Here, we examine the influence of out-planting coral restoration domes on the fish community over an 85-day period on an isolated, well-protected reef, in the Central Caribbean Sea.

## MATERIALS & METHODS

### Study site and data collection

We conducted our experiment on the remote, well-protected, and isolated reef of Little Cayman, situated within the Central Caribbean (Fig. 1A). Using metallic dome frames (1 m diameter), we attached coral fragments of *Acropora cervicornis* (Permit Ref. PSAP issued and signed by the Cayman Islands Department of Environment on behalf of the National Conservation Council) from the Central Caribbean Marine Institute coral nursery (*Maneval et al., 2021*). In total, we had five out-planting dome sites with coral fragments attached (Fig. 1B), each with three connected rebar frames to make one dome, covering an area of 3 m$^2$. All sites were situated between 18–21 m depth, with the location of sites haphazardly selected to avoid covering live, healthy coral colonies and separated by a minimum of 10 m. To quantify the change in the fish community in response to the out-planting of coral domes, we used a before-after experimental approach. While a before-after control-impact (BACI) approach would have been preferable (*Christie et al., 2019*), there was a dearth of control replicates for the BACI approach to be feasible. We performed fish surveys on each dome site beginning with an initial survey prior to placement of domes, five days after placement, and 85 days after placement. Each replicate was performed by a different individual for all the domes (that is, one dome survey has three replicates, performed by three different individuals) to account for surveyor bias. Surveys were conducted using the Stationary Point Count method (SPC) with all fish within an imaginary cylinder (2.5 m radius from the dome central point) from the benthos to the surface counted, and identified down to species level, with total length estimated to the nearest cm (*Samoilys & Carlos, 2000*). We also categorized Parrotfish into terminal and initial stages based on visual identification. Fish that left the vicinity of the sampling area and came back were not recorded twice if they were identifiable as the same individual. Surveying took place for 10 min, with a 2-minute acclimation period, with each survey triplicated to estimate averages for each dome. There was a minimum time delay of 3 min between each replicate to allow individual surveyors to swap domes. All surveys took place between 10 am and 2 pm on the 6th of April, 11th of April, and 30th of June, 2023. Fish species were subsequently grouped into their trophic guilds (herbivore, invertivore, macrocarnivore, omnivore, planktivore) based on dietary information derived from FishBase (*Froese & Pauly, 2010*), following (*Johnson, Chequer & Goodbody-Gringley,*

*2023*). After fish sizes were categorized *in situ* their biomass was calculated from size bins (0–5 cm, 6–10 cm, 11–20 cm, 21–30 cm, 31–40 cm, and >40 cm) using the formula:

$$W = a * L\hat{}b.$$

where W is the weight of the fish, L is the maximum length based on the size classes above, and a and b are species-specific constants based on empirical data for calculating fish biomass from size-weight relationships (*Bohnsack & Harper, 1988*; *Coull, 1989*; *Torres Jr, 1991*; *Kulbicki et al., 1993*). These constants were obtained from FishBase, with values from congeneric species used if data for a specific species were not available (*Froese & Pauly, 2010*). For fish that were larger than 40 cm, they were recorded to the nearest 10 cm interval, which was then used in the biomass calculation for that individual.

## Data analysis

To examine the influence of restoration domes on the fish community, we compared the mean biomass, mean abundance, and mean species richness of fish at each dome across the sampling period using non-parametric Kruskal-Wallis test, as data were not normally distributed based on visual inference of histograms and Shapiro–Wilks test of normality. For discerning the influence of restoration domes on the fish community, we used multivariate analysis to compare the community composition of reef fishes before out-planting, five days after out-planting, 85 days after out-planting using non-Metric Multi-Dimensional Scaling (nMDS). We implemented the nMDS using the 'Vegan' package (*Oksanen et al., 2022*), where data were square root transformed before implementing a Bray-Curtis dissimilarity transformation on the community matrix (*Oksanen et al., 2022*). We then compared dissimilarity between the communities using a PERMANOVA (*Oksanen et al., 2022*).

## RESULTS

We found no significant differences in the biomass ($\chi^2 = 1.82$, $df = 2$, $P = 0.403$), abundance ($\chi^2 = 5.469$, $df = 2$, $P = 0.065$), or species richness ($\chi^2 = 1.007$, $df = 2$, $P = 0.605$) of all fish from before out-planting, compared to day 5, and day 85 since out-planting (Fig. 2).

This pattern was consistent for functional groups and selected taxa, with no significant differences in abundance or biomass of fish before out-planting, five days after out-planting, and 85 days after out-planting (Table 1, Fig. 3).

When comparing the community composition of reef fish before out-planting compared to 85 days after out-planting of restoration domes, there was no significant difference in the community composition (Fig. 4, PERMANOVA, $df = 2$, $F = 1.324$, Sum of Squares $= 0.198$, $P = 0.143$). Additionally, the dominant species were generally consistent in their abundance at the outplant sites across sampling periods (Fig. 5).

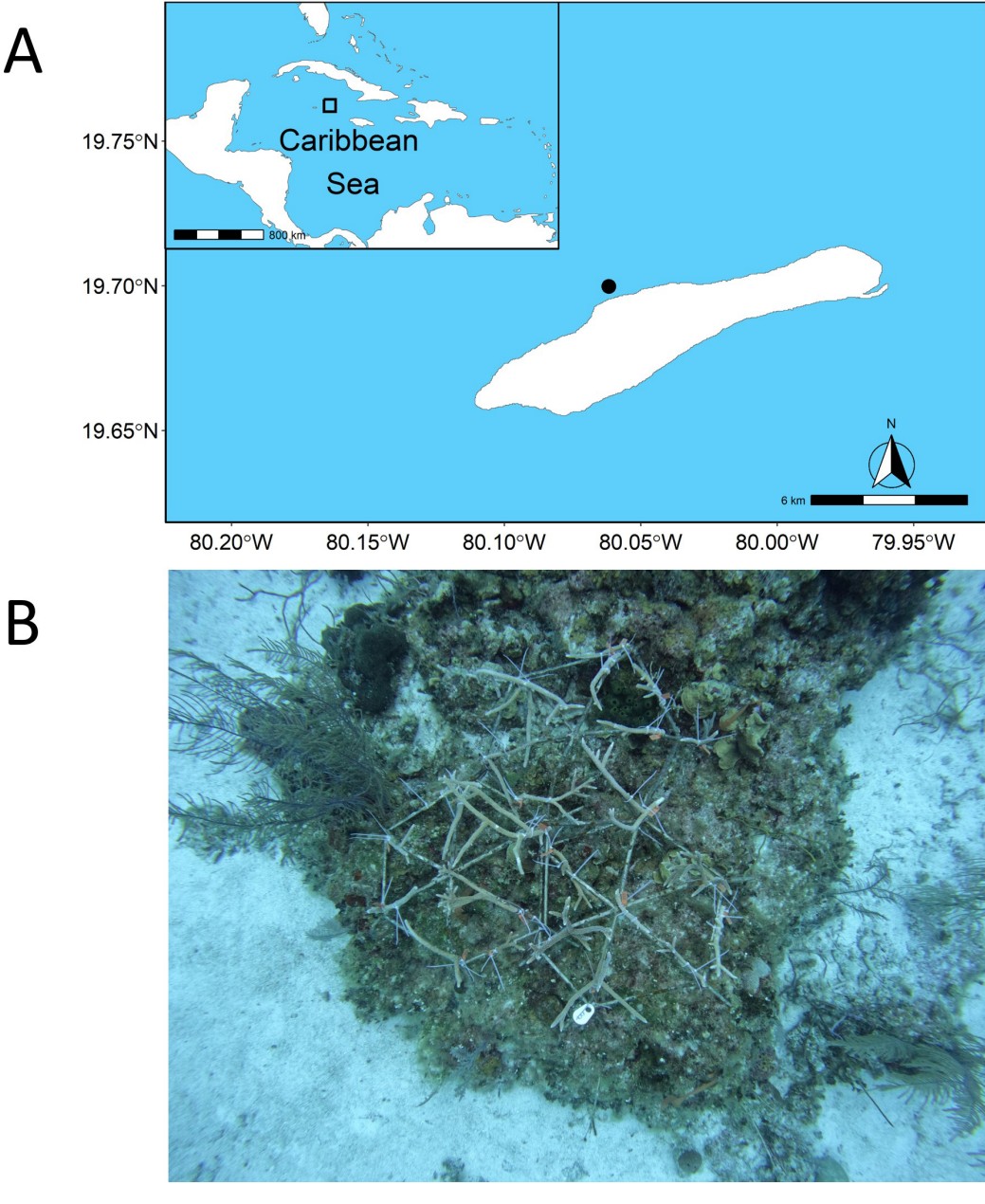

**Figure 1** **Site map and example of experimental design.** Location of the study site where restoration domes were out planted in Little Cayman shown within the Caribbean Sea (A). An example of one restoration dome outplant site is shown in (B) on day zero of out-planting. Site map generated from the package "rnaturalearthdata" (https://cran.r-project.org/package=rnaturalearth). Photo credit: Alex Chequer.

## DISCUSSION

Our findings highlight no short-term effect of restoration domes for enhancing the biomass, abundance, or species richness of reef fishes in Little Cayman. This finding was consistent for functionally important fish groups. Additionally, the lack of shift in the fish community

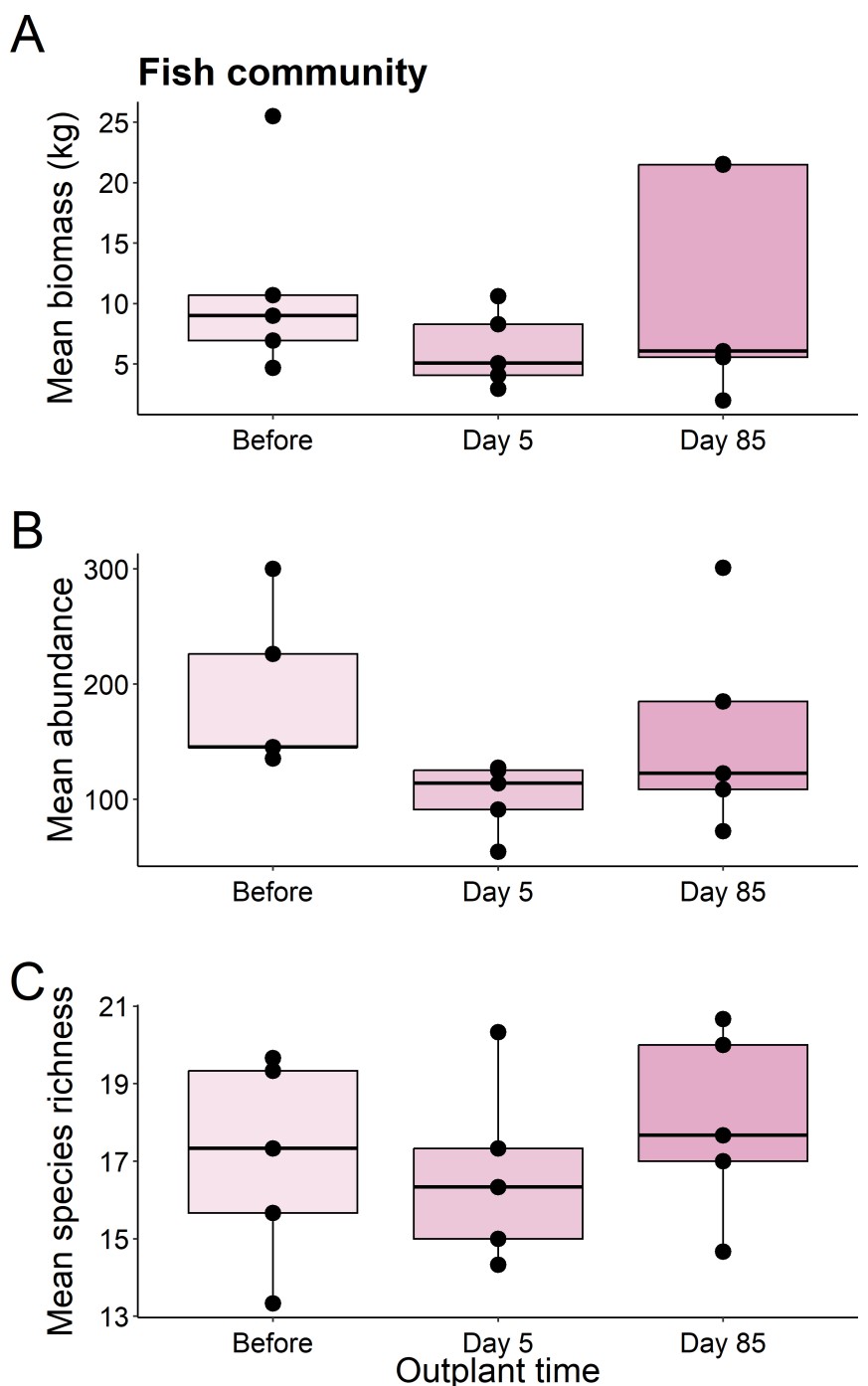

**Figure 2  Summary boxplots of fish community structure over the study period.** Plots show the Biomass (A), abundance (B), and species richness (C) for reef fish at the restoration dome out-plants across the sampling period. Boxes represent the first and third interquartile, whiskers show the range of the data calculated as 1.5 times the interquartile, horizontal bar represents the medium, and dots indicate outliers.

**Table 1  Kruskal-Wallis comparisons of functional groups and fish taxa.** Comparisons are changes in the fish biomass and abundance for before out-planting, five days after out-planting, and 85 days after out-planting.

| Functional group/taxa | Measure | $\chi^2$ | df | *P*-value |
|---|---|---|---|---|
| Herbivores | Biomass | 0.74 | 2 | 0.691 |
| | Abundance | 0.316 | 2 | 0.854 |
| All Parrotfish | Biomass | 0.38 | 2 | 0.827 |
| | Abundance | 0.06 | 2 | 0.97 |
| Initial Parrotfish | Biomass | 0.32 | 2 | 0.852 |
| | Abundance | 2.624 | 2 | 0.269 |
| Damselfish | Biomass | 3.92 | 2 | 0.141 |
| | Abundance | 0.622 | 2 | 0.733 |
| Macrocarnivores | Biomass | 1.86 | 2 | 0.395 |
| | Abundance | 0.925 | 2 | 0.63 |
| Omnivores | Biomass | 3.84 | 2 | 0.145 |
| | Abundance | 3.712 | 2 | 0.156 |
| Invertivores | Biomass | 3.14 | 2 | 0.208 |
| | Abundance | 4.645 | 2 | 0.098 |
| Planktivores | Biomass | 1.94 | 2 | 0.379 |
| | Abundance | 0.456 | 2 | 0.796 |

before out-planting compared to 85 days after out-planting suggests negligible influence of short-term coral restoration on the fish community in our isolated and well-protected reefs in Little Cayman.

The lack of influence from restoration domes on the fish community over our 85-day study period is unsurprising given the plethora of factors that affect reef fish communities. Within the Caribbean shifts in the community composition of fish associated with restoration either happened before drastic changes in reef function and composition since the turn of the century (*Hudson et al., 1989*) or are strikingly rare (*Opel et al., 2017*; *Seraphim et al., 2020*). Because local conditions including food availability (*Sale, 1977*), habitat complexity (*Gratwicke & Speight, 2005*), depth (*Pinheiro et al., 2023*), and direct anthropogenic pressures upon the seascape (*Exton et al., 2019*; *Duarte et al., 2020*; *Johnson, Chequer & Goodbody-Gringley, 2023*) are significant drivers of fish community composition, the influence of small-scale restoration domes is unlikely to elicit strong effects consistently. Any qualitative changes observed in the high abundance of species at one sampling period, for example, the high *Caranx latus* abundance at day 85 compared to previous sampling periods, can be attributed to the schooling nature of these fish.

Additionally, using restoration domes on a well-protected and isolated reef in Little Cayman will likely exert a strong influence on our findings. Up to July 2023 when this study ended, coral cover and structural complexity remained stable in Little Cayman, generally higher than the rest of the Caribbean region (*Goodbody-Gringley & Manfrino, 2020*). Fish populations have also remained stable, with high abundances, biomass, and species richness associated with the isolation from local impacts such as overfishing, and a network of marine protected areas around Little Cayman (*Goodbody-Gringley & Manfrino,*

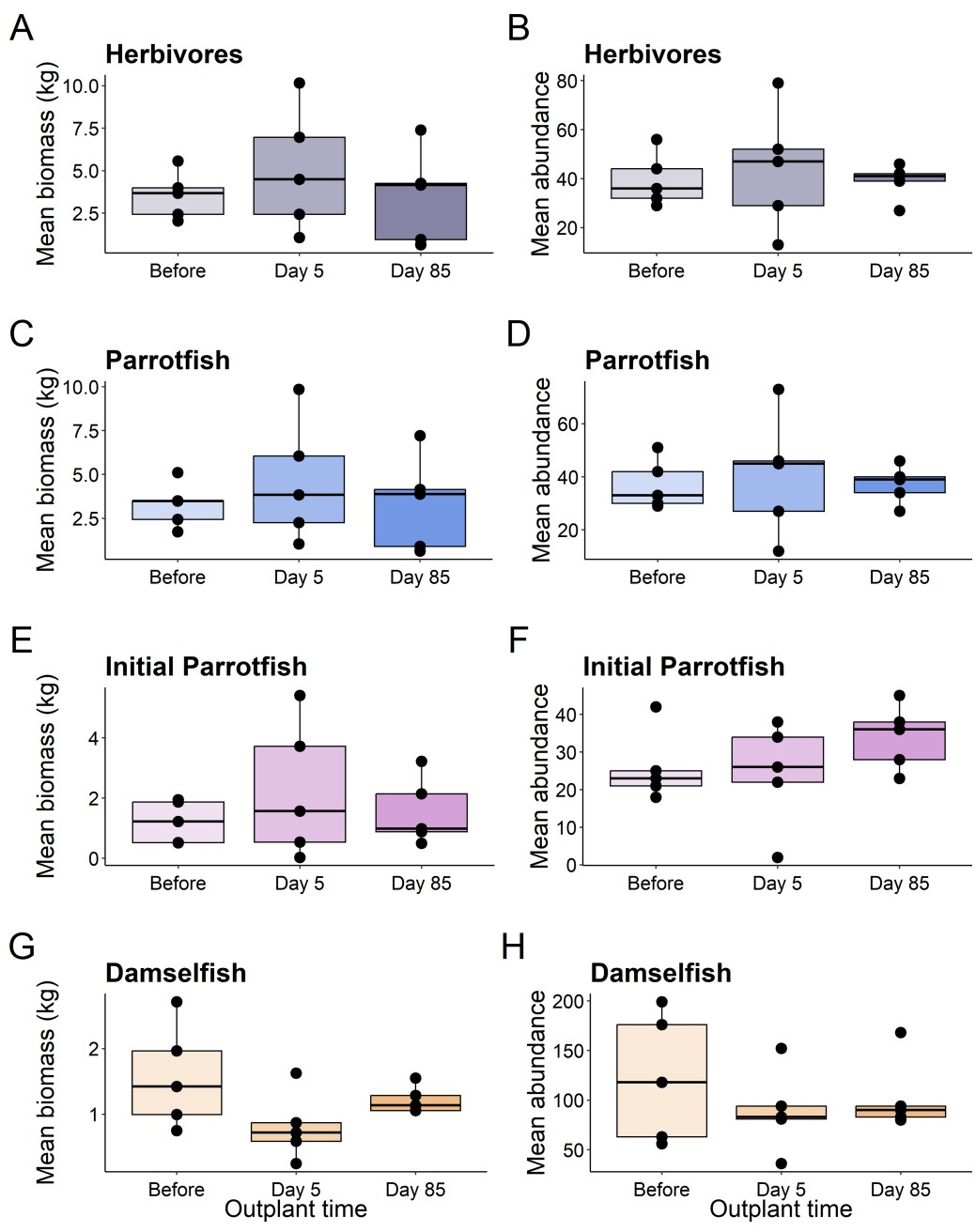

**Figure 3  Boxplots of selected reef fish functional guilds.** Plots show the biomass (left) and abundance (right) for selected reef fish functional groups at the out-planted domes across the sampling period. (A–B) are herbivores, (C–D) are Parrotfish, (E–F) are initial stage Parrotfish, and (G–H) are Damselfish. Boxes represent the first and third interquartile, whiskers show the range of the data calculated as 1.5 times the interquartile, horizontal bar represents the medium, and dots indicate outliers.

2020). Therefore, it is possible the fish community cannot be enhanced by a small-scale restoration project given the already underlying habitat complexity and stability of fish community structure. However, it should be noted that the addition of habitat complexity

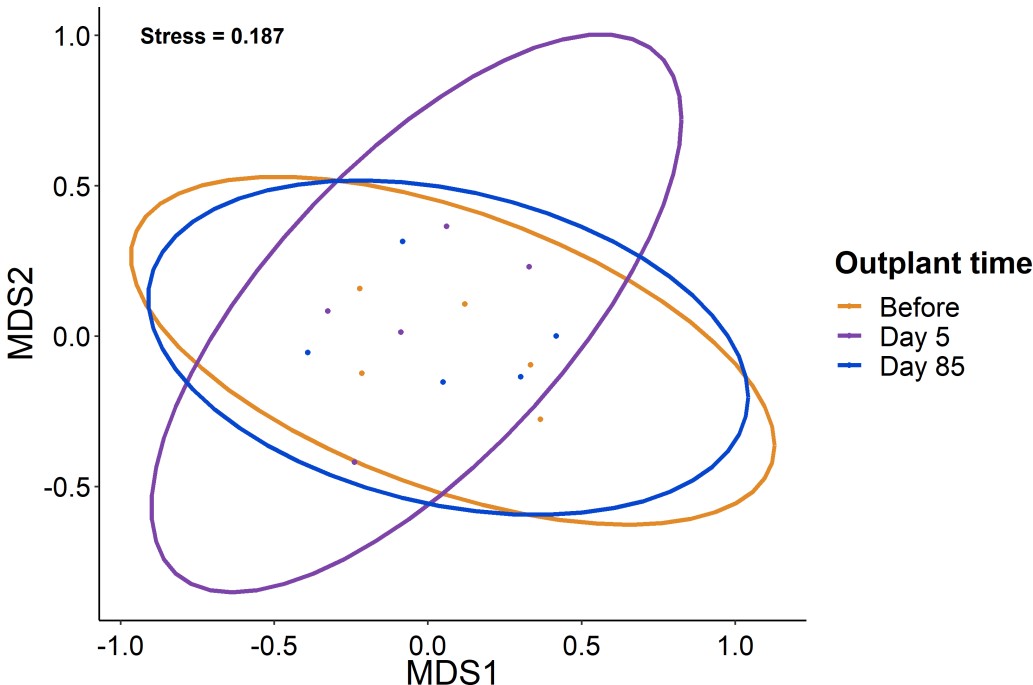

**Figure 4** **Lack of change in the fish community over the study period.** Ordination of the fish community from sampling before, five days after out-planting, and 85 days after out-planting. Points represent surveys while the ellipses constrain the entirety of the ordination space.

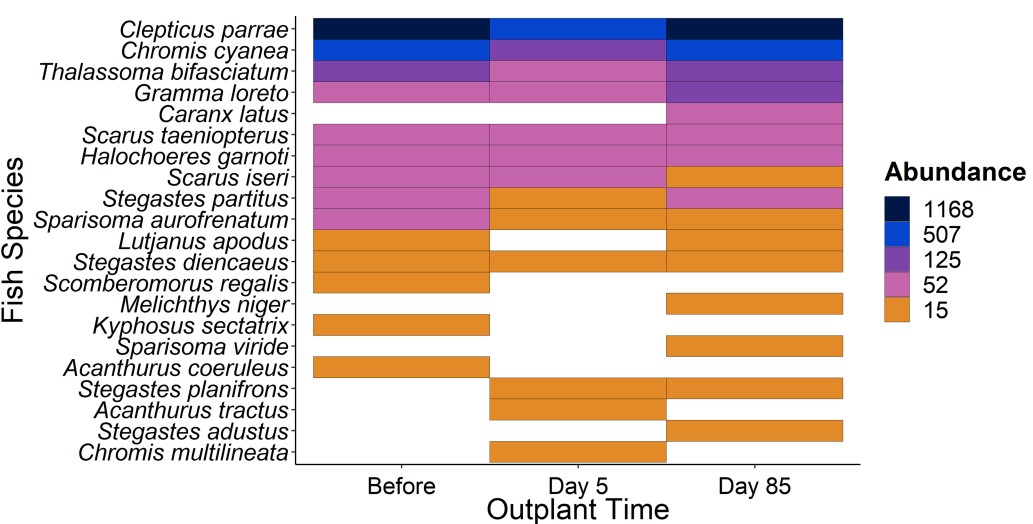

**Figure 5** **Overview of fish species during the study period.** Heat map of fish species abundance from the 21 most abundant species recorded from outplant sites before out-planting, five days after, and 85 days after out-planting of restoration domes.

has size specific effects on fish abundance (*Hixon & Beets, 1989*), as smaller sized fish utilize shelter provided by structural complexity (*Bohnsack et al., 1994*; *Nash et al., 2013*). Qualitatively, our species-specific analyses indicate that small fish species did increase in abundance (Fig. 5) during the study (*i.e., Gramma loreto, Stegastes adustus*, initial stage Scarids), yet this pattern did not exist for the small, highly residential pomacentrids (Fig. 3H).

However, given the negligible effect of our restoration domes on the reef fish community, our findings indicate restoration is unlikely to influence the fish community when conducted over a small spatial scale over 85 days. As other studies over larger spatial and longer temporal scales also consistently find restoration does not influence reef fish communities in the Caribbean (*Huntington et al., 2017*; *Ladd et al., 2018*; *Hein et al., 2020*; *Seraphim et al., 2020*), it is unlikely that enhancements of fish communities *via* coral restoration will be a regularly achieved goal, albeit with exceptions (*Huntington et al., 2017*; *Opel et al., 2017*). Thus, using reef restoration to enhance reef resilience through ecosystem processes will likely be extremely difficult to achieve (*Seraphim et al., 2020*; *Shaver et al., 2022*; *Hughes et al., 2023*). Rather, management strategies to control direct impacts of local stressors to reefs and reef fishes are likely far more important for reefs and fishes (*Hughes et al., 2017*), especially under global climate change (*Bruno, Côté & Toth, 2019*; *Eddy et al., 2021*). Yet, considering our study site is located on an isolated and well protected reef, where local stressors known to influence fish communities are reduced (*Manfrino et al., 2013*), our findings suggest even sites managed to enhance fish biomass are unlikely to show changes in the fish community as a response to coral restoration efforts. Given coral reefs are being annihilated by global climate change, and local-scale efforts cannot ameliorate resistance to warming (*Johnson, Dick & Pincheira-Donoso, 2022a*; *Johnson, Dick & Pincheira-Donoso, 2022b*) or generally enhance recovery (*Bruno & Valdivia, 2016*; *Cox et al., 2017*; *Bruno, Côté & Toth, 2019*; *Baumann et al., 2022*), the goalposts of what is achievable through restoration are shifting (*Hughes et al., 2023*). Perhaps within the Caribbean, even trying to influence fish communities through restoration is no longer achievable in the Anthropocene—at least for the majority of coral restoration projects, which tend to be short term because of funding and coral mortality after out-planting (*Hughes et al., 2023*). Rather, other direct management interventions such as fisheries regulations and effective management are far more likely to have ecologically relevant positive impacts on fish biomass and abundance (*e.g., Duarte et al., 2020*).

## CONCLUSIONS

In conclusion, we provide a Caribbean case study where out-planting of complex coral restoration structures did not influence the reef fish community on an isolated and highly protected coral reef. Our findings highlight the difficulty of using restoration to restore fish communities to enhance reef resilience *via* ecosystem function processes. However, our study covered a small spatial scale over an 85 days but is generally consistent with recent Caribbean studies (*Seraphim et al., 2020*). For these reasons, we speculate restoring corals in the Caribbean, where coral mortality is high (*Hughes et al., 2023*), is unlikely to

influence the fish community, and thus provides implications for coral resilience. Future research could focus on a longer-term study over a larger spatial scale to provide more detailed insights from a well-protected isolated reef. However, with continued rising ocean temperatures and marine heatwaves, restored and juvenile corals are continuing to be annihilated (*Lohr et al., 2017*; *Hughes et al., 2019*), making such endeavors increasingly difficult (*Hein et al., 2020*; *Boström-Einarsson et al., 2020*; *Shaver et al., 2022*; *Hughes et al., 2023*). Our findings back up the overwhelming evidence that restoring coral reefs and maintaining ecosystem function requires immediate reductions in greenhouse gas emissions to thwart the trajectory of global climate change and its impact on coral reefs.

## ACKNOWLEDGEMENTS

We would like to thank Victoria Mann, Leeav Cohen, Janna Randle, Lowell Forbes, and Haley Davis for field assistance and data collection. The authors are grateful to Rose Griffith for advice on statistical analysis. We are especially grateful the editor of this manuscript Guilherme Corte for handing our paper and finding three expert reviewers. We are thankful for the reviewer comments by Abel Valdivia, Juan Pablo Quimbayo, and one anonymous reviewer.

### Funding

This project was funded by a RESEMBID grant from the European Union, Dart, Stuarts Humphries, The Disney Conservation Fund, Cayman Water, The AALL Foundation Trust and The Ernest Kleinwort Charitable Trust. There was no additional external funding received for this study. The funders had no role in study design, data collection and analysis, decision to publish, or preparation of the manuscript.

### Grant Disclosures

The following grant information was disclosed by the authors:
A RESEMBID grant from the European Union, Dart, Stuarts Humphries.
The Disney Conservation Fund, Cayman Water.
The AALL Foundation Trust and The Ernest Kleinwort Charitable Trust.

### Competing Interests

John F. Bruno is an Academic Editor for PeerJ.

### Author Contributions

- Jack V. Johnson performed the experiments, analyzed the data, prepared figures and/or tables, authored or reviewed drafts of the article, and approved the final draft.
- John F. Bruno conceived and designed the experiments, performed the experiments, authored or reviewed drafts of the article, and approved the final draft.
- Lucas Le Gall performed the experiments, authored or reviewed drafts of the article, and approved the final draft.

- Matthew Louis Doherty performed the experiments, authored or reviewed drafts of the article, and approved the final draft.
- Alex Chequer conceived and designed the experiments, performed the experiments, authored or reviewed drafts of the article, and approved the final draft.
- Gretchen Goodbody Gringley conceived and designed the experiments, performed the experiments, analyzed the data, authored or reviewed drafts of the article, and approved the final draft.

## Data Availability

The data and codes are available at GitHub: Available at https://github.com/JackVJohnson/Coral-restoration-fish-community-Little-Cayman.

The data are also available at Zenodo: JackVJohnson. (2024). JackVJohnson/Coral-restoration-fish-community-Little-Cayman: Coral-restoration-fish-community-Little-Cayman (Version v1). Zenodo. Available at https://doi.org/10.5281/zenodo.10844688.

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
