# Peer review of "Creation of complex reef structures through coral restoration does not affect associated fish populations on a remote, well-protected, Caribbean reef"

_PeerJ, doi:10.7717/peerj.17855_

## Round 0.1 · original submission · Minor Revisions

Dear Dr. Johnson,

Your paper has been reviewed by three experts in the field. They all agree that your research was well-executed and your manuscript provides relevant information for coral restoration strategies. They also provided suggestions that I hope you address before your manuscript is accepted for publication. Please make sure to acknowledge their valuable contribution to the revised version.

Thank you for your submission to PeerJ.

·

Basic reporting

This manuscript explores the potential effects of coral restoration efforts on the associated reef fish community on an isolated and well-protected reef of Little Cayman Island in the Central Caribbean. The study investigates whether out-planting Acropora cervicornis colonies on raised domes influences fish biomass, abundance, and species richness. The results indicate no significant changes in fish community metrics following restoration efforts, suggesting a limited impact on fish populations in the study area. The authors discuss the factors influencing reef fish communities and interpret the findings in the context of the study's location and experiment duration. It concludes that small spatial-scale restoration initiatives do not enhance fish communities in isolated and well-protected reefs over short timescales.

In general, the manuscript adheres to professional language standards, provides relevant background information, references pertinent literature, and includes relevant figures with clear labeling and descriptions. The raw data was not presented.

Experimental design

The experimental design was well-conceived and executed from site selection, restoration, and surveying methods to data analysis. I commend the authors for choosing a remote, well-protected, and isolated reef as a controlled environment for the experiment, minimizing potential confounding variables. I also liked the use of metallic dome frames to attach coral fragments as they provide a more tridimensional structure that could potentially attract small-size fish. Unfortunately, there were only five restoration domes, limiting the power of the study. An addition of a control site(s) (with no domes) surveyed at the same time intervals would have been a more powerful design to detect impacts. The data analysis and the use of non-parametric statistical tests are appropriate for the non-normally distributed data. However, it's essential to consider potential limitations in the study, such as the relatively short duration of the study.

Validity of the findings

Overall, the findings presented in the study appear valid, supported by a sound experimental design, appropriate statistical analysis, and thoughtful interpretation of the results in the context of reef ecology and conservation. The findings in the study suggest that the out-planting of restoration domes had no significant influence on the biomass, abundance, or species richness of reef fish, nor did it result in significant changes in the community composition of reef fish compared to before out-planting. Similar results have been found in other studies. Although the authors appropriately acknowledge the complexity of reef fish communities and used an appropriate sampling design and data analysis, the study would have been more powerful on a longer time scale. However, the authors acknowledge the limitations of their study, including its relatively small spatial and temporal scale. They suggest that similar results have been observed in other studies conducted over larger spatial and longer temporal scales in the Caribbean. They also highlight the need for further research to confirm these findings and explore the potential for restoration to influence fish communities in other contexts.

Additional comments

I have suggested several detailed edits on the pdf regarding grammar and style (please see the attached document)

Please see below other general comments that are not included in the pdf.

Lines 83-85. Consider rewriting the idea to improve readability to something like: “Coral reef restoration ranged from small scale clearly defined projects (refs) to region-wide ambitious efforts (refs)”

Line 150: Clarify whether the total length was estimated to the nearest 1 cm.

Line 153: If each survey was performed three times on each dome (set of three), did you estimate averages for that dome?

Line 201: Add here that no effect on species richness was also observed

Line 218: I recommend starting a new paragraph here: “Additionally, using restoration domes…”

Line 226: In addition to larger spatial scales and longer time frames that may be needed to detect the impacts of coral reef restoration on fish assemblage structure, the first effects may be detected in smaller-size fish. I am wondering, if you compare small fish (1-10 cm TL) to larger fish (10-30cm TL) you see any difference in abundance and biomass. Previous studies have shown that artificial reefs have effects on fish abundance depending on size (see Hixon & Beets 1989). Smaller size fish tend to be associated with structures that provide shelter (see Bohnsack et al 1994, Nash et al 2012). Based on your species-specific data small fish species such as Gramma loreto and Stegastes adustus may be more abundant on day 85 than before out-planting corals (see Fig. 5).

Lines 242-248. I would reinforce here that large-scale and longer timeframes may be needed to see any measurable and ecologically significant effects of coral reef restoration on fish community structure. Perhaps highlight that reef restoration is another intervention in the portfolio of actions that need to be taken to stop and reverse the decline of coral reefs in the Caribbean in the long term. However, reef restoration may provide a much lower return on investment in reef fish biomass and abundance than other more direct management interventions like fisheries regulations and effective management.

References:

Hixon & Beets 1989 - https://www.researchgate.net/profile/Mark-Hixon-2/publication/216900219_Shelter_Characteristics_and_Caribbean_Fish_Assemblages_Experiments_with_Artificial_Reefs/links/5c64a98ca6fdccb608c11122/Shelter-Characteristics-and-Caribbean-Fish-Assemblages-Experiments-with-Artificial-Reefs.pdf

Bohnsack et al 1994 - https://www.researchgate.net/profile/James-Bohnsack/publication/233497940_Effects_of_Reef_Size_on_Colonization_and_Assemblage_Structure_of_Fishes_at_Artificial_Reefs_Off_Southeastern_Florida_USA/links/02e7e530b56b216f1c000000/Effects-of-Reef-Size-on-Colonization-and-Assemblage-Structure-of-Fishes-at-Artificial-Reefs-Off-Southeastern-Florida-USA.pdf

Nash et al 2012 - https://www.researchgate.net/profile/Kirsty-Nash-3/publication/243055545_Cross-scale_Habitat_Structure_Drives_Fish_Body_Size_Distributions_on_Coral_Reefs/links/02e7e51db41a1b7af4000000/Cross-scale-Habitat-Structure-Drives-Fish-Body-Size-Distributions-on-Coral-Reefs.pdf

Reviewer 2 ·

Basic reporting

no comment

Experimental design

Statistical analyses were sound. Experimental design may be suspect if the spatial and temporal scale of restoration/follow up do not match well with expected impacts on fish communities.

Validity of the findings

No comment

Additional comments

see attached comments

Annotated reviews are not available for download in order to protect the identity of reviewers who chose to remain anonymous.

·

Basic reporting

This manuscript explores the impact of coral restoration on fish communities in protected reefs in the Caribbean Region. This study is of broad interest to ecologists since it explores ecological processes that modulate local fish communities. Overall, this manuscript is well-written, appropriately uses the literature, correctly explores the dataset, and presents results and discussions consistent with the initial objective. Thus, I consider this manuscript relevant for framing a paper for this journal. However, I have some comments that I believe can further improve the quality of this manuscript.


Abstract

L47-53. The authors mentioned that no significant difference was observed in the fish community after 85 days of out-planting. However, it is not clear how this result was found. I suggest including a brief description of the statistical methods considered.

Introduction

L94. At the end of the first sentence ‘…coral reef ecosystems’, I suggest to adding some references on regarding the outcomes of coral reef restoration.

L127-129. Considering the mixed findings of restoration on fish communities, it might be interesting to add a general hypothesis on the expect in Cayman Island.

Material and methods

L147-150. I suggest adding the range of time intervals between Stationary Point Count Surveys, as the time between each cylinder can change and affect the fish community's metrics (i.e., species richness, abundance, and biomass).

L154-155. I suggest that you improve a description of the trophic guilds consider in your study.

L169-174. Although I believe that the authors used a correct statistical method, this method may not be the most robust and appropriate considering the distribution of each fish community metric. For instance, GLM with gamma distribution is more suitable for fish biomass, while GLM with Poisson is more appropriate for fish abundance.

L169-174. Additionally, I suggest that the authors add supplementary figures showing the residuals distribution of each model. Furthermore, they should test whether these values exhibit spatial autocorrelation using the Moran Index.

Results

L185-186. The authors presented results on fish community metrics for Parrotfish, but it was not clear how they classified between adult and juvenile Parrotfish. Thus, I suggest adding more details in the method section regarding this classification.

Discussion

The authors discussed their results in comparison with those observed in other regions, which was great. However, I wonder how certain areas can influence fish community metrics. For instance, small areas may not necessarily have a strong influence on habitat complexity, resulting in a lack of effect. I suggest that the authors explore more about the impact of the restoration area on fish communities in light of Island Biogeography Theory.

Experimental design

Not comments

Validity of the findings

Not comments

Additional comments

Fig 2. I suggest adding the points in the boxplot and letters that show that no were observed differences in the three different times (i.e., before, Day 5 and Day85)
Fig. 3. Same suggestion figure 2

---

## Round 0.2 · Minor Revisions

Dear Dr. Johson,

We have received three reviews of your manuscript. Overall, the reviewers are positive about your study and recommend its publication, but a few additional changes are still requested by reviewer 2. I also added my evaluation with minor comments (see below). At this point, my decision is “acceptable after revision.”

The main concern of the reviewers relies on the small spatial and temporal scale of your study and the absence of proper control treatment. I agree with them and acknowledge that you addressed the lack of control treatment in your response to reviewers. However, since this is a main issue in experimental design and has been constantly (and correctly) highlighted by the reviewers, we can expect that many readers of your work will also note it. Therefore, I suggest including some explanation in your manuscript to address the future audience (maybe a few sentences in the data analyses section).

I also acknowledge that you addressed the spatial and temporal limitations of your study; however, I believe you can tone down some of your statements in the Discussion section. For example, you start by saying, “Our findings highlight no effect of restoration domes for enhancing the biomass, abundance, or species richness of reef fishes”. Maybe it would be best to clarify that your results highlight no short-term effect of restoration domes (same comment on L213 - The lack of influence from restoration domes…).

The final paragraph of the Discussion section is well-balanced and makes it clear that “restoration is unlikely to influence the fish community when conducted over a small scale over an 85-day period”. You also did a great job showing that longer studies found similar results and pointing out that other management strategies may be more effective than restoration practices.

Finally, I ask you to carefully consider reviewer 2 comments, including reviewing your manuscript for clarity and completeness.

Thank you for your great work!

·

Basic reporting

This is the second time reviewing this manuscript, see my basic reporting summary in my first review. The authors have satisfactorily addressed all my comments and suggestions. No additional comments

Experimental design

See my previous experimental design comments in the first review. No additional new comments in this version.

Validity of the findings

See my assessment of the validity of the findings in my first review. No additional new comments here.

Additional comments

I thank the authors for reviewing and responding to my comments and suggestions line by line. I read the manuscript again and I have no further comments. I recommend reviewing the document again during the pre-print process to catch any typos (if any) I have missed. Well done.

Reviewer 2 ·

Basic reporting

Figures could use some sprucing up to be publication quality. Some unclear sections in the text (see pdf for specific line numbers).

Experimental design

Missing proper controls for a BACI design. Methods insufficiently described.

Validity of the findings

Lack of inclusion of observer effects in statistical model (or poorly described methods).

Additional comments

See PDF for specific comments

Annotated reviews are not available for download in order to protect the identity of reviewers who chose to remain anonymous.

·

Basic reporting

I appreciate the reorganization of the manuscript and the testing of whether GLMs are the most appropriate analysis manuscript. The authors have addressed all comments/suggestions, thus improving significantly the manuscript. I believe that this manuscript is ready for publication.

Experimental design

The authors have clarified all aspects related to sampling and sample processing in this
revised version.

Validity of the findings

The data and analysis support the findings/conclusions made by the authors. In this
revised version, the discussion is much clearer.

---

## Round 0.3 · accepted · Accept

Dear authors,

Thank you for accommodating all reviewers' suggestions and comments. I appreciate your manuscript and the effort you put into reviewing it. Great job!